# *Say Something, Do Something*: Evaluating a Forum Theater Production to Activate Youth Violence Prevention Strategies in Schools

**DOI:** 10.3390/ijerph21010039

**Published:** 2023-12-27

**Authors:** Keon L. Gilbert, Elizabeth A. Baker, Karen Bain, Julia Flood, John Wolbers

**Affiliations:** 1The Brookings Institution, Department of Behavioral Science and Health Education, College for Public Health and Social Justice, Saint Louis University, St. Louis, MO 63103, USA; 2Metro Theater Company, St. Louis, MO 63103, USA; 3Prison Performing Arts, St. Louis, MO 63103, USA

**Keywords:** youth violence prevention, health equity, arts and public health, program evaluation, community based research approaches, social-emotional learning

## Abstract

Background: Youth violence that takes place within school settings exposes youth to serious social, mental and physical consequences that affect education performance, and life opportunities. Previous work shows positive youth development frameworks can promote social-emotional learning by enhancing empathy and building problem-solving and conflict management skills. Theater-based interventions have been shown to enhance social emotional development by privileging youth voices, and building youth capacities and strengths. The current manuscript presents the evaluation of an arts-based and public health framework conducted to assess the development, implementation and impact of a forum theater production, Say Something, Do Something (SSDS) in St. Louis, Missouri. Methods: An iterative mixed methods approach was used, starting with observations of productions. Using convenience sampling, we then conducted post interviews of the theater team (n = 8) and school personnel (n = 10). Results: Respondents highlighted that as a result of engagement of school personnel in program development, the language and scenarios presented were relevant to students. Data indicated that SSDS increased student knowledge and changed attitudes, developed student conflict management and problem-solving skills, and improved interpersonal behavior. SSDS also raised awareness of the importance of, and created the foundation for, additional system and policy changes in the schools. Conclusion and implications: Forum theater is an approach that can enhance socio-emotional learning and conflict management among youth. Collaborative initiatives between public health and the arts are poised to uniquely engage community partners, animate interventions, and impact critical public health issues including youth violence prevention.

## 1. Introduction

Behaviors such as fighting, threats, and bullying are forms of youth violence that commonly take place within school settings [1,2]. Nationally, approximately 20% of 12–18-year-old students report being bullied on school property in the last year, and 9% indicate that on at least 1 day of the last 30, they did not go to school because they felt unsafe at school or on their way to or from school [3].

Public health has a key role in addressing youth violence by developing prevention strategies to address these issues. In order to do so effectively, we need to understand the social, structural, and proximate determinants contributing to youth violence. The various pathways and environments that both expose youths to and protect them from violence are important to know. Some have argued, for example, that youth violence is a byproduct of youth experiencing acute and chronic stressors stemming from systemic oppression, racism, and living in impoverished neighborhoods [4]. Exposure to these life stressors and traumatic experiences, or adverse childhood experiences (ACEs), can occur across key developmental periods.

These experiences can lead to the onset of chronic diseases, decrease socio-economic opportunities, and further expose those in these communities to additional stressors and traumatic experiences, including violence within community and school settings. Exposure to youth violence has serious consequences in terms of injuries, increases in risky health behaviors, poor mental health, low educational achievement, and an increased risk of further engagement in violence [5].

## 2. Enhancing Social–Emotional Learning as a Way to Build Resilience and Reduce Youth Violence

Recent reviews have found that when addressing youth violence, it is important to address a continuum of violence prevention, from disrespectful comments to physical aggression, and in doing so, to focus less on changing attitudes and punishment and more on skill development and capacity building [6,7,8]. Suggestions include addressing antecedent behaviors and providing young people with opportunities for intrapersonal reflection and practice in interpersonal conflict management. Previous work indicates that changing peer norms and the school climate around bullying are critical to reducing school violence [9]. Positive youth development and resilience frameworks that promote social-emotional learning provide youths with healthy social and problem-solving skills, such as enhanced empathy, problem solving, and conflict management. This, in turn, can reduce the risk of violent behaviors, improve social skills, boost educational achievement, and improve job prospects [5,10]. Other data suggest positive impacts not only on the youths who learn these skills but on their peers as well. Data indicate that more than half (57%) of bullying situations stop when a peer intervenes on behalf of the student being bullied [11]. These interventions have been found to be most effective when they focus on elementary- and middle-school youths, engage teachers and staff to help build these skills, and provide opportunities for parents to learn anti-violence strategies [4].

## 3. Role of Schools in Addressing Youth Violence

Community stakeholders who are interested in reducing youth violence may turn to schools to help youth respond to structural inequities in ways that build resilience so that they are better able to manage stressors and excel both within and outside the school environment. We have to consider the social and physical environments that contain school spaces. These spaces influence school-based experiences, exposures within school decisions young people routinely encounter. Data from the National Center for Education Statistics showed that there are more schools in high-poverty areas than in low-poverty areas. The racial and ethnic differences in who attends high-poverty schools are stark: 45% percent of Black students attend high-poverty schools, followed by 43% of Hispanic students, 37% of American Indian/Alaskan Native students, 25% of Pacific Islander students, 14% of Asian students, and 8% of White students [12].

Schools are a setting in which school-aged youth spend a vast amount of time. As a result, communities and families often look to schools to assist with youth development. Schools may similarly see the importance of addressing youth violence because of its effects on school-related factors, such as attendance and academic performance, or the resulting health effects, such as the mental health of students. However, school personnel, families, and communities may have different expectations and norms with regard to how to prepare youths to respond to conflict. While schools can assist youths in developing positive responses to conflict, they must do so while acknowledging that youths may have pressures to respond in different ways.

Another challenge that schools face is that they are often called upon to respond to violence while or after it occurs. In doing so, they may respond in detrimental ways, such as implementing zero-tolerance and other policies that bring youths in contact with the criminal justice system. Doing so often further exposes youths to additional trauma and facilitates the school-to-prison pipeline, rather than laying the foundation for positive development.

In order to address these challenges, school teachers and staff can enhance their capacity to intervene by bolstering resilience through the development of compensatory (direct) and/or buffering (indirect or interactive) protective mechanisms to address the effects of violence on youths. This requires both professional development and families, schools, and policy makers to work collaboratively with students to create sustainable change. As the field of public health continues to search for the best way to engage in these collaborations, integration of the arts has come to be seen as an important approach [13].

The arts have been shown to have a positive impact on youth development, including the enhancement of social-emotional development, in part by fostering opportunities for youths to voice their own realities, build capacities, and recognize their own strengths [10]. In particular, studies have found that engaging youths in participatory performing arts programs, such as forum theater, can build social-emotional skills, including problem solving, self-regulation, social competence, and enhanced resiliency [10]. Forum theater, also known as participatory theater or ethnodrama, is grounded in Paulo Freire’s empowerment theory [14]. Auguso Boal’s initial *Theater of the Oppressed* [15] work aimed to transform oppressive structures by leveraging the knowledge of those who have been disenfranchised. In general, these approaches invite audiences to become participants in the performance by asking them to envision ways to transform their current social realities. Audiences first engage with performers and/or facilitators to represent or reflect the reality of their current experiences and then co-create ways to modify these conditions using theater techniques to envision change [15,16]. Previous work has found that forum theater can be a useful tool for addressing social issues in the classroom and for violence prevention in general [17,18].

This manuscript describes the development and implementation of *Say Something, Do Something*, a forum theater production, and assesses its impact using an arts-based and public health framework adapted from Animating Democracy.

## 4. Context and Community Response to Youth Violence: St. Louis ReCAST (Resiliency in Communities after Stress and Trauma) Project

In 2016, the Saint Louis County Department of Health, in partnership with the City of St Louis Health Department and the Saint Louis Mental Health Board, began their ReCAST initiative in the designated Promise Zones [19]. The St. Louis Promise Zone is characterized by students in 4th–6th grade who live in areas where 99% of students receive free or reduced-price lunches, 15% have an IEP (Individualized Education Program), and 91% are African American, with 5% White, 1% Latinx, 1% Asian American, and 1% other. ReCAST engaged Promise Zone residents (over the age of 11) in a participatory process of reviewing and awarding micro-grants in a number of areas, including youth violence prevention. One of the projects they funded was Metro Theater Company’s *Say Something, Do Something*, which uses forum theater to build problem-solving skills to address school-based youth violence.

## 5. Participatory Theater for Social Change and Violence Prevention

### 5.1. Program Model Development: Say Something, Do Something

*Say Something, Do Something* is a signature program of Metro Theater Company in St. Louis, MO, USA (MTC). MTC is a non-profit theater company, motivated by the intelligence and emotional wisdom of young people. MTC comprises actors, teaching artists, producers, technicians, writers, and arts administrators (e.g., education director, community engagement manager, artistic director, managing director). MTC’s artists and administrators choose and/or develop new productions and programs that help young people grapple with complicated social issues, such as racism, inequity, and physical or mental health concerns.

MTC worked with a local school district to develop *Building Community Through Drama* (*BCTD*), a five-part residency to help explore the triangle of oppression—bully, target, bystander, upstander, and collaborator (who helps the bully)—through role playing and reflective dialogue. Students make connections between historical and current instances of oppression. This program helped students to improve interpersonal communication to reduce the marginalization of other students as a result of the negative stereotypes that middle-school cultures engender. However, because of the large time commitment, *BCTD* was not seen as transferable to other school districts. When MTC was selected by RECAST, MTC engaged school-based stakeholders to identify the core components of the *BCTD* program and distill it into a 1 h production with a complementary educator resource guide. The production they created, *Say Something, Do Something*, uses participatory or forum theater, as described below.

### 5.2. Say Something, Do Something (SSDS) Production Includes Five Program Components

Introduction to theater: a facilitator introduces the idea of theater.Understanding body language and using thought bubbles to identify inner thoughts and encourage empathy: Students are encouraged to think empathetically through exercises in reading the body language of professional actors. Utilizing a cut-out thought bubble, students are asked to comment on how the actor is feeling based on their body movements, body posture, and facial expressions.SSDS scene: Students are then presented with a short scene incorporating situations in which a protagonist is being targeted by an antagonist or perpetrator. Taken from interviews with students, teachers, and counselors, the scene incorporates the language that students use daily and illustrates how these situations arise and escalate. Just prior to the point of conflict, the facilitator steps in to invite the students to help.Hot seating: Students interview the characters to find out why they made the choices they did.Students suggest ideas for a more positive outcome. By substituting themselves/tagging out the actors, students try out their ideas with the actors in a safe environment to be better prepared for when these situations occur in real life.

The entire production lasts approximately 50–60 min, with the SSDS scene itself lasting about 5 min. Prior to the production, teachers and staff are provided with educational materials that will help them to facilitate classroom conversations both prior to and after the production. *SSDS* is best seen as an educational workshop in which acting/theater helps support the learning process, as opposed to a touring show https://www.metroplays.org/blog-2022-2023/say-something-do-something (accessed on 17 November 2023).

### 5.3. Arts-Informed Evaluation Framework: Animating Democracy’s Continuum of Impact

While there are many publications articulating the benefits of performing arts for health [14,20,21], most evaluations of theater-based interventions have examined the satisfaction with and aesthetic quality of the production [16]. However, to adequately assess the potential impact of arts-based interventions to promote health, it is important to integrate arts-based frameworks and public health evaluation frameworks. We were also aware that using a public health framework alone would serve to privilege that lens over an arts-making lens, thereby limiting the lessons learned rather than maximizing the positive impact of MTC’s work [22]. Considering the assessment of the social and civic intent of the artistic production required us to incorporate aesthetic attributes, as defined by artists themselves. We had to reflect on the artistic production, its purpose, and its intended effects. The integration of these disciplines and models creates opportunities to assess the development (formative evaluation), implementation (process evaluation), and impact (impact and outcome evaluation) of the production on a public health issue. As a result, combining arts and public health evaluation frameworks expands the areas of evaluation and builds our capacity to reflect more deeply than using either framework alone.

Animating Democracy’s model for assessing the continuum of the impact of the arts was seen as useful for this type of evaluation [23] (see Figure 1). The framework was initially developed to provide evidence for the contribution of the arts to civic and social change [23,24]. Much like ecological models in the social sciences, it is important to think about change and impact across levels of determinants that can create an ecosystem to support better health outcomes. Animating Democracy helps us to think about a continuum of change within individuals in terms of awareness, beliefs, and knowledge; social engagement; and changes in physical conditions, systems, and structures [23] (see Figure 1). These impacts are seen as being achieved through the processes of engaging and reflecting, animating, informing, and impacting [23]. In addition to developing this continuum of impact, Animating Democracy developed a set of aesthetic attributes that contribute to the potential impact of the work. These include, for example, the extent to which the artistic endeavor demonstrates a commitment to the stated “cause” of the work (in this case, a reduction in school-based youth violence), disrupting or challenging existing norms and modeling new norms of responses, resourcefulness or the use of expertise, and the stickiness or evidence of potential sustainability. The application of this continuum can be achieved through the processes of *engaging and reflecting, animating, informing, and impacting* (see Figure 1).

Our evaluation began with evaluators observing ten productions. This provided the opportunity to gain a first-hand experience of the response to the production and some initial understanding of the factors influencing the implementation (location, number of students, age of students, etc.) and the overall aesthetics of the production [24]. Evaluators then reviewed the survey that Metro had developed to assess initial student responses to the production. The decision was made to build on MTC’s existing evaluation by conducting additional surveys and interviews with school teachers and staff to assess what was working well and what could be improved for future productions. The analysis of these data employed both traditional public health and arts-based frameworks to assess the development, implementation, and initial impact of the forum theater production *Say Something, Do Something* (*SSDS*).

## 6. Methods

*SSDS* was performed in 23 schools for a total of 2539 individuals, 2349 of whom were students who lived in the Promise Zone area (described above). The first stage of this evaluation was formative, which allowed the evaluation team to understand the structure and function of the *SSDS* production. This time was utilized to learn about the formation of the production, the training and preparation of MTC actors and facilitators, and the engagement of schools to prepare the production content from discussions with MTC staff and participant observations (n = 10 productions observed). The second stage of the evaluation was process and summative. Process and summative evaluation data were collected from interviews with the theater team (n = 8), school personnel (n = 10), and school personnel (n = 13) across participating schools. The main questions that these data helped to answer centered on immediate responses and reflections on the key messages delivered by the *SSDS* production, observations of short-term behavior changes in students, and recommendations about how *SSDS* can either become a frequent intervention across schools or how to build a sustainable theater-based youth violence initiative within individual schools.

### Data Analysis

The interviews were audio-taped and transcribed verbatim. The transcripts were then coded. Two individuals developed the initial codes using the Animating Democracy framework [23] and expanded as additional ideas were generated from interview data. These initial codes were collated, and sub-codes were developed as appropriate. A universal codebook with code definitions was developed. A team of two then coded each transcript to identify themes and quotes that best represented each theme. The themes were more fully described, reviewed, and modified as needed using a thematic analysis based on the Animating Democracy framework. The quotes with their corresponding themes are presented according to the program model. Reflections from participant observations and quotes from interviews correspond to or are aligned with how we learned about *SSDS* and its demonstration of the implementation and potential effect on students and school staff. Reporting our results in this way helps to communicate the alignment between our arts-based program development and our public health evaluation approach.

## 7. Results

### 7.1. Arts Engage: Integrating Students’ Lived Experiences

Metro Theater Company (MTC) began the development of *SSDS* by holding discussions with school leaders about some of the issues, challenges, behaviors, and language that students generally use in schools and experiences that may relate to youth violence. Many respondents described some of the issues or behaviors that were causing conflict, bullying, and possible violence within their schools:

Well, because our fifth grade classrooms were havin’ a pretty strugglin’ time with bullying and talkin’ about each other, so we felt like that’d be a good avenue for them to further discuss it as a class, what works, what doesn’t work, what should be appropriate and that.

They’re at the age group where they think it’s popular to be …makin’ fun of each other’s clothes. While some people might laugh, we just try to break it down to them how everybody doesn’t think it’s funny, and how easy it is for you to say, like, “I’m not your friend”, to one kid, or—from the minor stuff to the really, really big stuff, we’re just tryin’ to make sure that they understand how important it is to not fall into the bullying, or into the talkin’ about people, or into the gossiping and the rumors…

Along with bullying, a lotta kids don’t even understand what sexual harassment is or to what degree it is when they’re in school…At one point, some of the girls would go through, walk through, and they made this challenge of grabbin’ butts. A touchin’-butt challenge or somethin’. Other kids, they didn’t like it, but they didn’t know how to respond to it, either. That’s along the lines of *S[ay] Something, Say Something*, you know.

The above quotes help to illustrate that MTC’s process of engaging school leaders and teachers ensured that the production was credible and resonated with students’ lived experiences. Some of the schools that hosted the *SSDS* production had previous relationships and had hosted other programs produced by MTC:

When I heard about and I experienced Say Something, Do Something several years ago when they did it before, we really liked it, so when I heard they were doing it again this year—I received a call from them. They received a grant to provide it for some schools that might have students at risk—which we have plenty of, and they know that because they work with us all the time—so we were on their list to make sure that they brought it to us.

The continuity of programming within schools that have great need are important when developing arts-based programs and tailoring the context to their unique challenges. The facilitators, actors and school staff who were interviewed all emphasized that the students readily identified with the scenario presented in the production. The benefit of this process was noted across several interviews:

When Metro was actually writing the script and asked me some questions about what are the words that kids use, and what do they say to each other when their bullying each other, so I was hoping that our kids would recognize themselves… that they would hear themselves because the words in the script were some of their own words that kids use, like “You suck” was one of ‘em, and that’s what our kids say to each other quite often when they’re upset or angry or don’t like the way somebody kicked a ball, or whatever it may be. They use that ugliness. That’s how they criticize one another.

The play was right on target for the age group we serve. And then also [they] bring in the street…of where are kids come from. These things are really realistic that could be happening in their neighborhoods or in their homes. How they deal with it? We actually were talking about how it is good that the actors and actresses mostly looked like them, sounded like them, the children. And their uncles. And their aunts. And their parents. Some of our parents are about that age.

They had the same message, but even more so, the storyline for Say Something, Do Something was even more relatable to the kids and what’s happening right now. Some of the terminology that they used, how they acted. The kids were like, “I do that, too”. They were able to literally put themselves in that character. Then when it was time for them to actually play it, they could then change themselves in that moment.

### 7.2. Arts Animate: Production Implementation

The multiple school staff members who were interviewed were pleased and appreciated the interactive and engaging components of the performance.

Then it was interactive, so the students actually got to practice how to handle themselves in a conflict, which, you know, I don’t think we always get the practice part. We always get the information, but never the application opportunity.

I noticed that they were—well, for the main part, they were listenin’. They was really intrigued about what was goin’ on, and they had a lotta good input into the story and into the options as to how to fix it. Then they got a chance to actually see if their ideas worked. They were definitely very, very engaged. That part was pretty good to watch. Some of the kids who actually were strugglin’ with the bullies, and some of the ones who were actually doin’ the bullyin’, they all had good input onto how you should change it and how you can do better, so it worked out really well.

The respondents to the interviews noted that the engagement in the production (Hot Seating) was unique in that they saw students beginning to develop problem-solving and conflict management skills. They highlighted that these skills were developed when students were asked first to consider what each of the characters was experiencing and then to think through the situation and conflict and consider what the best course of action is from the perspectives of the different characters in the situation. The types of alternatives to physical conflict suggested by the students included seeking clarification on issues or concerns, intervening on the victims’ behalf, directly confronting the oppressor, walking away, or seeking assistance from someone else with power (an adult). The respondents saw this as potentially aiding in positive conflict resolution in the future. Exploring conflict resolution strategies was seen as beneficial in helping the students continue these practices after the play and follow through on these ideas when a new conflict arises in their lives.

Although these strategies are exploring different characters, this exercise is helping the students work through how different characters can resolve a conflict within the same space and that everyone has a role.

I saw some positive responses from students that I think had been bullying others in the past or had been bullied themselves. That they responded with “ok here’s what I should do” and they had the proper response I think.

This process of sitting and discussing conflict resolution through with student is important because it allows them to see that there’s two sides to every problem. Not just their side. And then just really stressing to them to try to solve things in a peaceful way. Try to keep peace. And if you can’t do that, you can always walk away. You can tell an adult.

Part of learning how to avoid conflict was developed through the *SSDS* methods used to help students with the process of building and understanding body language and empathy.

Like when [Metro performers] did the bubbles. I thought that was really helpful because it helped the kids to see in a situation, there are different sides to every story. And everyone is coming to it in their own mindset and their own chain of events could have happened before even that moment. And so to kind of take a step back and think about maybe how the other person feels and have empathy for that person and think about that. So, I thought that was really good. Because usually, it just helps them to see that there are more ways to look at a problem and there’s two sides to every problem. Not just their side. So I thought that was good.

There was an instance where somebody was tryin’ to spread rumors, but instead of feeding into the rumors, the kids that actually—when they brought the rumor to one of the kids, the kid went quickly to the person who the rumor was about, let them know so they may work together without even havin’ to add the teachers, or add any other people on, or getting’ into it even further. They made it a conversation to where they can figure out where the rumor started and how it got misunderstood. They pretty much worked it out. The teacher was in the class. She didn’t say anything, but she let it play out, and she actually watched, watched them fix it, watched them handle it.

For some schools, *SSDS* was a new and unique approach to structuring reflective discussions with students about challenges within their homes, in their neighborhood environments, and with other students. For others, it was seen as reinforcing other activities and policies that they already have in place.

The teachers, my fellow teachers and I, we have policies in place we have disciplinary procedures in place. What I most wanted to get out of this was the students being able to recognize those situations themselves. And then be able to, either de-escalate or use different tactics that would achieve a better outcome than, say, getting suspended for fighting or actually being injured physically. Or emotionally injured. So if the effect of SSDS was that we have a few more tools in our toolbox when we’re faced with a bullying situation, that would be my ideal outcome.

Across various *SSDS* performances, teachers, counselors, and other staff were engaged to talk about their role in addressing school violence. Teachers were called upon to ask them what they would do in certain situations when conflict arises or someone comes to them to report bullying. The respondents to the interviews indicated that the teachers often seemed to be at a loss for strategies to address these issues.

There’s a part where we ask them, teachers what do you do in these kinds of situations. And 9 times out of 10, it feels like it’s crickets. Nobody wants to answer… the teachers just look at each other… like who’s going to throw themselves on their sword to answer this question.

### 7.3. Arts Influence: Awareness, Knowledge, Empathy, Skill Building, and Policy Change

Teachers, counselors, and principals were also asked to indicate whether they had seen any behavior changes in the students who attended the production. Eight of the thirteen respondents indicated that they had seen changes and provided specific examples. These included having seen the students specifically discussing the content of the production, reflecting on previous experiences of bullying and fighting in light of what they had seen in the production, and using the strategies and techniques learned in the production to resolve conflicts that arise.

Fighting is a big problem in our school. The message presented on how to avoid fights…instead use their words was especially appreciated. Thank you for helping to enforce what we try so hard to teach them every day!

I see the kids discussing the play and heard them reflecting on previous situations involving conflict, bullying, and fighting. I hope the conversation continues and the strategies they saw are used.

Yes. A few days after the performance a 7th grade student who is well known for having issues with other students helped to resolve a conflict between two of his classmates.

Not only did students seem able to understand the messages of *SSDS*, but they also had initial positive reactions, showing some immediate behavior changes. According to school staff and those in daily contact with the students, a “positive same day response” was noted.

There was an incident about somebody’s dad was in jail that had happened here at the school. And one of the kids told everybody. So it really sets the mood when someone was talking about someone’s mom in the play. And the resolution was that, you know, things happen. People have circumstances. You never know tomorrow what is going to be for you. And the girl [girl at the school] that went and told everyone about the other girl’s father came back and apologized. All along she had been denying it. ‘I didn’t say that. I didn’t say that!’ But she came back and apologized to her friend. I don’t know if it’s the play that did it. I had the counselor talk to her *before* the play and she said ‘I didn’t say such a thing’. But she did come back *after* the play and apologized to her friend.

I can’t really say that we stopped using certain words, but I do think that that word gay’, it just comes up too much as an insult, and they used in the—and I told John that at the beginning. I said, ‘That really is disturbing to me’, and I can’t ever find the right words to help the kids understand that there are so many—if you have to insult somebody, there are other ways. They did put that in the script, and our kids did—I do think that they were shocked when they heard somebody else saying it ‘cause it was so awful when they said that to that guy in the performance. Maybe that ‘cause you can feel it—their reaction. I can’t say that I’ve even heard that insult since, but I don’t know if it’s a direct result of seeing it in the play.

There was an instance where somebody was tryin’ to spread rumors, but instead of feeding into the rumors, the kids that actually—when they brought the rumor to one of the kids, the kid went quickly to the person who the rumor was about, let them know so they may work together without even havin’ to add the teachers, or add any other people on, or getting into it even further. They made it a conversation to where they can figure out where the rumor started and how it got misunderstood. They pretty much worked it out. The teacher was in the class. She didn’t say anything, but she let it play out, and she actually watched, watched them fix it, watched them handle it.

Sustaining this, using the messages and strategies, and getting closer to a school-culture change were noted as important goals across respondents. Some requested resources. Those who asked for additional resources to help reinforce take-away messages identified ways to integrate these within their curriculum or to create opportunities to prepare students before the production and to reinforce messages and strategies after the production.

[W]hen we first did this, I felt like we were going to schools where the soil was already tilled and raked. We’re just planting the seeds and preparing the soil. And this time it felt like we were just going in and just dropping the seeds and hoping that they stick. Because…the preparation for to prepare for the arrival for this information. We’re here. One day. Drop this on you. And we’re gone. That’s what it felt like this time. That felt different … This time there wasn’t much follow-through that we can do because we don’t have a presence always in the classroom as well. It’s…it kind of does feel a little bit like ‘hello, goodbye’ almost.

I think maybe some websites or some other places that we could go to get more information about how to deal with situations such as that [in the performance]. Or even just send us a survey that we could send to parents and kinda gauge their responses. In previous programs where I’ve worked with, sometimes the company will hand out written surveys to the students. And they [students] can work with them [surveys] and the teachers can collect them. And either send the data back to metro or give it for our purposes or both. If that makes sense.

*Many school staff members expressed a desire for more post-production engagement for teachers and others at the schools*. A few of the schools noted that the production was part of a broader conversation that the school community was having. As such, it set the stage for future discussions.

[T]hink the performance was a catalyst or the start of the conversation and it’s like an ongoing conversation that we’re having with kids. And so it brought it up in a way that was real for our kids. But it also allowed then for our teachers to then transition back to their class and have a real conversation with kids. One that wasn’t forced. And it also…it would have a conversation about something that happened in the production but it felt very real. Because it wasn’t something that had actually happened at (our school), there was no blame or we weren’t talking about one of our kids. But we were talking about something that was very real for our kids, if that makes sense.

More engagement with parents was identified by some respondents as a way to better understand and reinforce the student’s response to the performance.

I think METRO may be able to build something like this into a more parent-directed, you know, force for this play. Because what was going on was in the school building. When kids come home and tell their parents that somebody say something to them. What do our parents really do? What is their response to that?

Several respondents noted that additional resources would be beneficial when applied to developing problem-solving and conflict resolution strategies with students, school teachers, counselors, administrators, and families.

## 8. Discussion

Arts-based interventions in public health are needed to engage youths to build their social–emotional development and to build conflict resolution skills to prevent violence within and outside of school contexts. Our evaluation highlighted engagement strategies to reach schools struggling with violence within their schools and create programs that can address the trauma that youths cope with as a result of their social contexts. Arts-based programs like *SSDS* do not create imaginary scenarios for students to rehearse. Forum theater is not just a unique form of drama for the audience but a valuable intervention strategy to engage school communities, which include students, teachers, school staff, families, and the broader community. The use of forum theater has the capacity to challenge the norms of youth relationships and enhance their social ties by using language, increasing respect for their differences, and learning how to support their peers who may have challenges. The skills that students were introduced to were practical, easy to rehearse, and relevant to many scenarios or situations that they had encountered. This was the result of the *SSDS* production content utilizing their language and behaviors. Forum theater provided a new learning context for students. In traditional theater productions, actors draw a production to a logical conclusion when ending the last scene. In forum theater, which is guided by a skilled facilitator, the audience and actors co-create the ending, which can improve the adoption of new skills and behaviors, as observed in the results of our evaluation.

Our evaluation strategy integrated arts-and-health promotion evaluation models into a framework that may be useful for others searching for ways to understand the effects of arts-based projects focusing on a range of social issues, health behaviors, and social determinants of health. Using an arts-based and public health evaluation framework facilitated our knowledge about how this production, as a model, served as an entry point for longer-term arts-based violence prevention efforts within school settings. Forum theater, as an application of participatory theater, is part of a continuum of community engagement strategies using drama to promote health and well-being. This is an important engagement strategy within public health that can help youths recognize and develop their own strengths in managing challenges that they encounter.

Participatory theater also provides a unique opportunity for sustainable partnerships between the arts, public health, and school communities. A key strategy that MTC employed was to engage school communities to inform the intervention by using language, the style of dress, and contexts that would resonate with youths. While working in communities such as the Promise Zone, identifying issues, strengths, and assets informs a more holistic intervention that creates a safe space for youths to explore coping skills to remain safe within their schools. These efforts can sustain interest, resources, and the capacity to adapt *SSDS* and other arts-based programs to address current public health issues and trends.

## 9. Limitations

During the intervention, several program implementation lessons were learned. It was not always possible for productions to be facilitated as intended given time constraints; in some schools, there were concerns because there were too many students attending or the space where the production took place was not conducive to student engagement, such as gymnasiums or other open spaces. There were also challenges when older students attended along with middle-school students. Future productions will need to attend to these logistical challenges and realities. One suggestion is to tailor arts-based health promotion strategies to the audience with unique message strategies and the application of specific forms of art. Our evaluation also highlights that the impact of *SSDS* on youth behavior would be enhanced if there were additional opportunities to engage parents and other members of youth social networks.

Additional limitations include not being able to engage youths directly. The start of our evaluation began after the start of the *SSDS* production. Evaluators developed a practical and rapid strategy to learn about the potential impact of *SSDS* to aid MTC’s additional engagement with schools to host productions and possibly create longer-term residencies to dive deeper into violence prevention strategies. Due to time constraints and the inability to obtain consent from parents and assent from students, we had a convenience sample of teachers, school counselors, and school administrators. We were not able to interview students to capture their feedback about *SSDS* and how they were integrating the messages, lessons, and skills they learned. The current iteration of *SSDS* was not able to provide as many pre/post-production resources to assist teachers and school staff as they desired. Providing these additional resources as part of professional development within schools may be an important part of future arts-based programs that are conducted within schools.

## 10. Conclusions

Accounting for these limitations, *SSDS* demonstrated great promise as an arts-based health promotion program that can be disseminated and implemented on a larger scale. Our evaluation highlighted engagement strategies to reach schools struggling with violence within their schools and the trauma that youths cope with as a result of their social contexts. Arts-based programs like *SSDS* do not create imaginary scenarios for students to rehearse. Rather, by grounding themselves in students’ lived realities, these programs can arm students with simple strategies that may prevent violent incidents and suspension and promote safety and wellness within schools and within their home communities.

## Figures and Tables

**Figure 1 ijerph-21-00039-f001:**
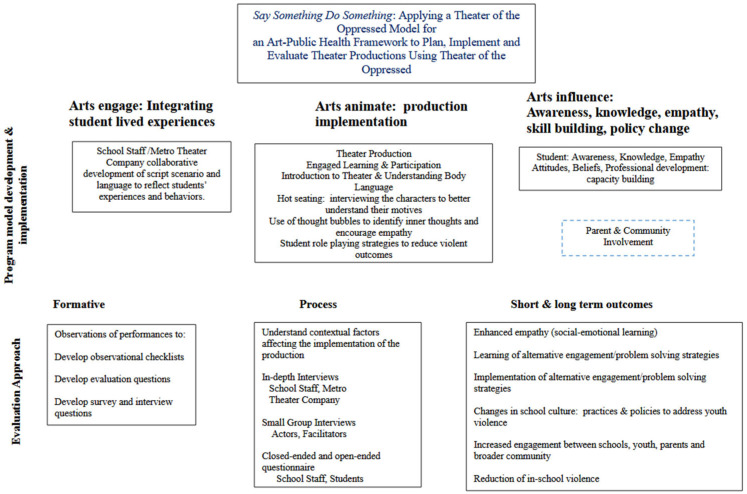
Arts-based program model and evaluation approach. Adapted from: Animating Democracy. How do arts & culture make a difference? Continuum of Impact: A guide to defining social & civic outcomes & indicators, Americans for the Arts.

## Data Availability

The data are not publicly available due to privacy considerations.

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
