# Peer review of "Say Something, Do Something: Evaluating a Forum Theater Production to Activate Youth Violence Prevention Strategies in Schools"

_ijerph, 2023, doi:10.3390/ijerph21010039_

Round 1

Reviewer 1 Report

Comments and Suggestions for Authors

All the comments in attached file

Author Response

Please kindly check the attached document

Reviewer 2 Report

Comments and Suggestions for Authors

Given the well-known concept of the school-to-prison pipeline, preventing youth violence at schools is important. In this regard, by evaluating an art-based program, this manuscript offers an important insight. However, I see there are problems that need to be addressed before publication. 

First, the authors should provide a more comprehensive background about youth violence. In particular, they should discuss what prevention measure is currently being taken and why it is failing to prevent youth violence in order to highlight the potential contribution of the theater production. 

I cannot seem to find Figure 1. Is it the one on Page 4? It is unclear, as there is no caption. Also, how does the figure on Page 5 differ from the attachment named "Making Meaning of the Arts Based Intervention and Evaluation Final Model"? Relatedly, the authors should explain this model in more detail. What are the relationships between each concept in the model?

It is not clear how the themes were identified based on the data collected. The authors should provide this information. It is also not clear why the authors present the corresponding codes in the form of playbills. An explanation is needed. 

Results are not presented in a coherent manner, such that it is difficult to follow how these findings help to answer the research question of this manuscript. 

The discussion section is very thin, despite the richness of the data and results. The authors should provide more discussions about how this project contributes to preventing youth violence in schools.

Author Response

(The authors gave the same response as above.)

Round 2

Reviewer 2 Report

Comments and Suggestions for Authors

The quality of the manuscript has significantly improved. However, the authors need to use consistent formatting in the Results section. L Some quotes lack a quotation mark. Long quotes need to be indented. 

Author Response

The quality of the manuscript has significantly improved. However, the authors need to use consistent formatting in the Results section. L Some quotes lack a quotation mark. Long quotes need to be indented. 

From Lead Author: I have gone through the results section and double-indented longer quotes. Within several quotes the person speaking to us during the interview references something another person said. Those references to the words of other people are indicated or offset with a single quotation mark. There are other punctuation marks that indicate their use of a shortened or slang word such as [gettin'] the single quotation mark tells readers a letter is left off from the speaker. 
